# GLP-1 receptor agonists' impact on cardio-renal outcomes and mortality in T2D with acute kidney disease

Heng-Chih Pan[1,2,3,4,5], Jui-Yi Chen [6,7], Hsing-Yu Chen[8,9,10], Fang-Yu Yeh[11], Chiao-Yin Sun[3,5], Thomas Tao-Min Huang[11,12] & Vin-Cent Wu [11,12] ✉

Previous studies have explored the effects of glucagon-like peptide-1 receptor agonists (GLP-1 RAs) in reducing cardiovascular events in type 2 diabetes. Here we show that GLP-1 RAs are associated with lower risks of mortality, major cardiovascular events (MACEs), and major adverse kidney events (MAKEs) in type 2 diabetes patients with acute kidney disease (AKD). Utilizing global data from the TriNetX database (2002/09/01-2022/12/01) and propensity score matching, we compare 7511 GLP-1 RAs users to non-users among 165,860 AKD patients. The most common causes of AKI are sepsis (55.2%) and cardiorenal syndrome (34.2%). After a median follow-up of 2.3 years, GLP-1 RAs users exhibit reduced risks of mortality (adjusted hazard ratio [aHR]: 0.57), MACEs (aHR: 0.88), and MAKEs (aHR: 0.73). External validation in a multicenter dataset of 1245 type 2 diabetes patients with AKD supports the favorable outcomes. These results emphasize the potential of GLP-1 RAs in individualized treatment for this population.

Type 2 diabetes is a global epidemic and a known independent risk factor for AKI and subsequent decline in kidney function[1–3]. In various healthcare settings, acute kidney disease (AKD) has seen a rising incidence[4,5], with associated heightened risks of all-cause mortality, end-stage kidney disease, and chronic kidney disease (CKD)[6]. Notably, type 2 diabetes often accelerates kidney function decline even before the onset of AKI, emphasizing the combined influence on CKD development[7–10]. Managing of type 2 diabetes within the context of AKD poses distinctive challenges demanding innovative solutions to alleviate the global CKD burden.

Glucagon-like peptide-1 receptor agonists (GLP-1RAs) present a promising avenue for addressing the complex cardiovascular-kidney-metabolic health[11,12]. They stimulate insulin secretion, inhibit glucagon release, and slow gastric emptying, potentially exerting protective effects on renal function[13]. Previous trials have demonstrated that GLP-1 RAs can reduce the risk of major adverse cardiovascular events (MACEs) and may have beneficial effects on kidney function[14–17]. Notably, the American Diabetes Association advocates for the utilization of GLP-1 RAs with established cardiovascular benefits in patients diagnosed with type 2 diabetes and concurrent cardiovascular disease. While the studies supporting these benefits were not initially focused on kidney health and included patients with low kidney risk, they suggest a link between GLP-1 RAs treatment and kidney protection, especially for type 2 diabetes patients with CKD[18].

[1]Graduate Institute of Clinical Medicine, College of Medicine, National Taiwan University, Taipei, Taiwan. [2]Chang Gung University College of Medicine, Taoyuan, Taiwan. [3]Division of Nephrology, Department of Internal Medicine, Keelung Chang Gung Memorial Hospital, Keelung, Taiwan. [4]Community Medicine Research Center, Keelung Chang Gung Memorial Hospital, Keelung, Taiwan. [5]Kidney Research Center and Department of Nephrology, Linkou Chang Gung Memorial Hospital, Taoyuan, Taiwan. [6]Division of Nephrology, Department of Internal Medicine, Chi Mei Medical Center, Tainan, Taiwan. [7]Department of Health and Nutrition, Chia Nan University of Pharmacy and Science, Tainan, Taiwan. [8]Graduate Institute of Clinical Medical Sciences, College of Medicine, Chang Gung University, Taoyuan, Taiwan. [9]Division of Chinese Internal Medicine, Center for Traditional Chinese Medicine, Chang Gung Memorial Hospital, Taoyuan, Taiwan. [10]School of Traditional Chinese Medicine, College of Medicine, Chang Gung University, Taoyuan, Taiwan. [11]Division of Nephrology, Primary Aldosteronism Center of Internal Medicine, National Taiwan University Hospital, Taipei, Taiwan. [12]NSARF (National Taiwan University Hospital Study Group of ARF), and CAKS (Taiwan Consortium for Acute Kidney Injury and Renal Diseases), Taipei, Taiwan. ✉e-mail: q91421028@ntu.edu.tw

Building upon these findings, we conducted this longitudinal investigation using an extensive global medical records database to explore the associations between GLP-1 RAs and mortality, adverse cardiovascular and kidney-related events in a substantial cohort of individuals with type 2 diabetes concomitant with AKD. Our study aims to provide valuable real-world perspectives on the influence of GLP-1 RAs therapy among these patients.

## Results

### Study population characteristics

A total of 417,322 patients with AKI who required dialysis were discharged from the hospital. In the cohort of 165,860 AKD patients who could withdraw from acute dialysis, the mean age was 59.0 years, and 49.7% were male. We identified 7,511 individuals who used GLP-1 RAs and did not undergo dialysis or die within 3 months post-discharge were enrolled as the GLP-1 RAs users group (Table 1). Therefore, the prevalence of GLP-1 RAs users was 4.5% (7,511 of 165,860). The remaining 158,349 patients who did not use GLP-1 RAs were enrolled as the GLP-1 RAs non-users group. The median follow-up period for the entire cohort was 2.3 years. The 25th percentile indicates a duration of 1.2 years, while the 75th percentile extends to 3.5 years, and the 90th percentile reaches 4.3 years. Sepsis emerged as the most common cause of AKI in this study, accounting for 55.2% of cases, followed by cardiorenal syndrome at 34.2% (Table S1). Additionally, Table S2 contains information regarding eGFR and electrolyte levels after discontinuation of dialysis.

The average age of the GLP-1 RAs users group was 59.0 years, compared to 63.3 years in the non-users group. The proportions of female and Caucasian patients were similar in the two groups (50.3% and 65.1% vs. 46.5% and 66.2%, respectively). After PSM, 7492 GLP-1 RAs users were matched with an equal number of non-users (controls) for analysis (Fig. 1). Both groups exhibited insignificant differences in age, sex, race, comorbidities, medications, and most lab results. The mean eGFRs in the GLP-1 RAs users and non-users groups were 73.2 and 75.9 ml/min/1.73 m², respectively. Throughout the follow-up period, HbA1C levels remained consistently higher in the GLP-1 RAs group (Figure S1). In terms of kidney function, while the initial eGFR for GLP-1 RAs users was notably lower, the difference was no longer significant by the D60-90 interval, hinting at a possible stabilization or amelioration in kidney function attributed to GLP-1 RAs therapy (Figure S2).

### The impact of GLP-1 RAs on mortality, MACEs, and MAKEs

The crude incidence of mortality rate was 22.91 per 1000 person-years, which was significantly lower in the GLP-1 RAs users group than in the non-users group (6.8% vs. 12.9%; aHR: 0.57, 95% CI: 0.51–0.64) (Table 2, Table S3-4). The results indicated lower risks of MACEs (14.8% vs. 18.8%; aHR: 0.88, 95% CI: 0.80–0.96) and MAKEs (10.8% vs. 16.0%; aHR: 0.73, 95% CI: 0.66–0.80) and in the GLP-1 RAs users group (Table 2, Table S5-6), supporting the association between GLP-1 RAs use and improved outcomes in type 2 diabetes patients with AKD (Fig. 2). Our analysis indicates that there is no significant influence from unmeasured confounding variables (E-values for the point estimates [lower limits of the CI] were 2.90 [3.32], 1.54 [1.92], and 2.09 [2.39] for mortality, MACE, and MAKE, respectively). (Table S3)

### Subgroup, sensitivity and specificity analyses based on the treated population

Subgroup analyses were performed focusing on age, comorbidities such as hypertension and advanced CKD, and medication use (Fig. 3). The findings consistently indicated an association between the GLP-1 RAs use and a lower risk of mortality. An association between a lower risk of MACEs and the use of GLP-1 RAs persisted regardless of hypertension, advanced CKD, or insulin/metformin use.

This association was especially notable among younger patients without proteinuria and those not receiving sulfonylurea, dipeptidyl peptidase-4 inhibitors, or RAAS blockers.

Similarly, an association was observed between GLP-1 RAs use and a lower risk of MAKEs, although it was less pronounced in patients with proteinuria and those receiving short-acting GLP-1 RAs. To validate these findings, multiple sensitivity analyses were carried out using a variety of selection criteria and models that integrated various covariates (Table S7). These analyses consistently supported the primary analysis results. Remarkably, the sensitivity analysis for patients prescribed exendin-based GLP-1 RAs, contraindicated in cases of severe renal insufficiency (eGFR <30 ml/min/1.73m² according to current clinical recommendations)[19,20], also corroborated the primary analysis's results (Table S8-9).

Notably, a significant proportion (49.6%) of GLP-1 RAs users had already been on these medications prior to the index hospital discharge. Additional sensitivity analysis compared new GLP-1 RAs users with individuals commencing other second-line antihyperglycemic treatments, including Sulfonylureas, dipeptidyl peptidase-4 inhibitors, or Pioglitazone. The results of these analyses were consistent with the primary approach (Table S10). Furthermore, HbA1C levels in the GLP-1 RAs users were consistently higher than those in users of other second-line antihyperglycemic treatments throughout the study period, aligning with the primary analysis findings (Figure S3).

Specificity analyses also consistently indicated significant associations between GLP-1 RAs use and reduced risks in different composite adverse outcomes, further substantiating the robustness of our results (Figure S4).

### Positive and negative outcome analyses

We then performed positive outcome analyses, and found markedly increased rates of nausea after GLP-1 RAs treatment (aHR: 1.47, 95% CI 1.33-1.62) as a reported complication. To verify the robustness of our examination techniques, we further evaluated seven negative outcome controls, including conjunctivitis, melanoma, fracture, traffic accidents, osteosarcoma, lupus, and Crohn's disease. These conditions were not expected to have any connection with the use of GLP-1 RAs, and the outcomes revealed no meaningful associations between any of them and the use of GLP-1 RAs (Figure S4-5).

### External validation

We then corroborated our findings using data from 1,245 patients diagnosed with type 2 diabetes concomitant with AKD in the CGRD database. Forty-four of these 1,245 patients (3.5%) used GLP-1 RAs, and comparative analysis revealed that the GLP-1 RAs users group had remarkably reduced risks of MACEs (aHR: 0.48, 95% CI: 0.30–0.77, P = 0.002) and MAKEs (aHR: 0.39, 95% CI: 0.24–0.62, P < 0.001) compared to the nonusers group (Figure S6).

## Discussion

In this study, we observed that 12.9% of the patients with type 2 diabetes concomitant with AKD, who were not treated with GLP-1 RAs, experienced mortality within 5 years after discontinuing dialysis for AKI. In contrast, those treated with GLP-1 RAs showed significant reductions in the risks of mortality, MACEs, and MAKEs during a median follow-up period of 2.3 years. These findings are robust and supported by various predefined sensitivity tests and external data validation. The results of both positive and negative outcome analyses further strengthen our approach and underscore the potential benefits of GLP-1 RAs in different follow-up strategies across subgroup analysis and sensitivity tests. Given the rising prevalence of cardiorenal events in patients after AKD and the potential role of AKD in exacerbating these conditions, it is essential for healthcare providers to consider GLP-1 RAs treatment as an integral part of a comprehensive strategy to address this significant public health issue.

**Table 1 | Baseline characteristics of the study subjects before and after PSM**

| | Before matching | After matching | | | | |
|---|---|---|---|---|---|---|
| | GLP-1 RAs group (*n* = 7511) | Control group (*n* = 158349) | Std diff | GLP-1 RAs group (*n* = 7492) | Control group (*n* = 7492) | Std diff |
| **Demographics** | | | | | | |
| Age, mean ± SD | 59.0 ± 12.8 | 63.3 ± 15.0 | 0.309 | 59.0 ± 12.8 | 59.4 ± 14.3 | 0.024 |
| Male, *n* (%) | 3733 (49.7%) | 85825 (54.2%) | 0.089 | 3724 (49.7%) | 3746 (50.0%) | 0.006 |
| Female, *n* (%) | 3778 (50.3%) | 73632 (46.5%) | 0.078 | 3768 (50.3%) | 3746 (50.0%) | 0.006 |
| White, *n* (%) | 4890 (65.1%) | 104827 (66.2%) | 0.023 | 4877 (65.1%) | 4862 (64.9%) | 0.004 |
| Not Hispanic or Latino, *n* (%) | 5994 (79.8%) | 124304 (78.5%) | 0.033 | 5979 (79.8%) | 5979 (79.8%) | <0.001 |
| **Comorbidities, *n* (%)** | | | | | | |
| Hyperlipidemia | 4897 (65.2%) | 84083 (53.1%) | 0.248 | 4885 (65.2%) | 4450 (59.4%) | 0.120 |
| Chronic kidney disease | 2434 (32.4%) | 48296 (30.5%) | 0.041 | 2427 (32.4%) | 2015 (26.9%) | 0.121 |
| Proteinuria | 623 (8.3%) | 6967 (4.4%) | 0.161 | 622 (8.3%) | 427 (5.7%) | 0.102 |
| Congestive heart failure | 2088 (27.8%) | 45605 (28.8%) | 0.023 | 2083 (27.8%) | 2090 (27.9%) | 0.003 |
| Hyperuricemia | 75 (1.0%) | 950 (0.6%) | 0.045 | 75 (1.0%) | 45 (0.6%) | 0.044 |
| Ischemia heart diseases | 2862 (38.1%) | 58906 (37.2%) | 0.017 | 2854 (38.1%) | 2787 (37.2%) | 0.018 |
| Cerebrovascular diseases | 1217 (16.2%) | 28820 (18.2%) | 0.053 | 1214 (16.2%) | 1244 (16.6%) | 0.010 |
| COPD | 1157 (15.4%) | 24702 (15.6%) | 0.006 | 1154 (15.4%) | 1139 (15.2%) | 0.005 |
| Musculoskeletal disease | 5370 (71.5%) | 99126 (62.6%) | 0.189 | 5357 (71.5%) | 5432 (72.5%) | 0.022 |
| **Medications, *n* (%)** | | | | | | |
| Metformin | 4078 (54.3%) | 50513 (31.9%) | 0.465 | 4068 (54.3%) | 3986 (53.2%) | 0.021 |
| Sulfonylureas | 1743 (23.2%) | 25811 (16.3%) | 0.175 | 1738 (23.2%) | 1603 (21.4%) | 0.044 |
| DPP4i | 826 (11.0%) | 11084 (7.0%) | 0.140 | 824 (11.0%) | 787 (10.5%) | 0.015 |
| Acarbose | 23 (0.3%) | 158 (0.1%) | 0.039 | 22 (0.2%) | 7 (0.1%) | 0.016 |
| Insulin | 6722 (89.5%) | 113220 (71.5%) | 0.468 | 6705 (89.5%) | 6735 (89.9%) | 0.014 |
| Aspirin | 4214 (56.1%) | 76641 (48.4%) | 0.155 | 4203 (56.1%) | 4196 (56.0%) | 0.003 |
| Clopidogrel | 1645 (21.9%) | 19002 (12.0%) | 0.268 | 1641 (21.9%) | 1191 (15.9%) | 0.026 |
| Atorvastatin | 4214 (56.1%) | 62548 (39.5%) | 0.337 | 4203 (56.1%) | 4255 (56.8%) | 0.015 |
| Allopurinol | 526 (7.0%) | 10293 (6.5%) | 0.021 | 524 (7.0%) | 472 (6.3%) | 0.029 |
| Febuxostat | 30 (0.4%) | 475 (0.3%) | 0.014 | 30 (0.4%) | 22 (0.3%) | 0.013 |
| Alpha-blocker | 1089 (14.5%) | 22802 (14.4%) | 0.002 | 1086 (14.5%) | 1011 (13.5%) | 0.027 |
| Beta-blocker | 4522 (60.2%) | 86775 (54.8%) | 0.108 | 4510 (60.2%) | 4495 (60.0%) | 0.003 |
| CCB | 3162 (42.1%) | 59381 (37.5%) | 0.095 | 3154 (42.1%) | 3147 (42.0%) | 0.004 |
| ACEI or ARB | 5273 (70.2%) | 86300 (54.5%) | 0.329 | 5259 (70.2%) | 5334 (71.2%) | 0.022 |
| **Laboratory** | | | | | | |
| BMI | | | | | | |
| ≥30 kg/m$^2$ | 991 (13.2%) | 32145 (20.3%) | 0.191 | 989 (13.2%) | 989 (13.2%) | 0.002 |
| <30 kg/m$^2$ | 2261 (30.1%) | 35787 (22.6%) | 0.171 | 2263 (30.2%) | 2173 (29.0%) | 0.025 |
| Missing | 4259 (56.7%) | 90417 (57.1%) | 0.008 | 4240 (56.6%) | 4330 (57.8%) | 0.024 |
| SBP | 128.5 ± 19.0 | 128.0 ± 20.5 | 0.026 | 128.5 ± 19.0 | 127.3 ± 20.0 | 0.061 |
| WBC, x10$^3$/uL | 9.1 ± 64.6 | 10.3 ± 95.6 | 0.015 | 9.1 ± 64.6 | 9.9 ± 82.6 | 0.011 |
| Platelet, x10$^3$/uL | 253.4 ± 97.5 | 246.3 ± 106.8 | 0.069 | 253.4 ± 97.5 | 256.7 ± 107.4 | 0.032 |
| eGFR, mL/min/1.73m$^2$ | 73.2 ± 31.8 | 71.7 ± 35.6 | 0.045 | 73.2 ± 31.8 | 75.9 ± 35.4 | 0.081 |
| Proteinuria, mg/g | 40.1 ± 36.1 | 43.0 ± 37.9 | 0.078 | 40.1 ± 36.1 | 42.6 ± 39.5 | 0.066 |
| Total cholesterol, mg/dL | 163.5 ± 53.2 | 162.1 ± 58.4 | 0.025 | 163.5 ± 53.2 | 165.6 ± 63.0 | 0.037 |
| HbA1C | | | | | | |
| ≥ 7.0% | 2276 (30.3%) | 60014 (37.9%) | 0.162 | 2270 (30.3%) | 2225 (29.7%) | 0.013 |
| <7.0% or missing | 5235 (70.7%) | 98335 (62.1%) | 0.534 | 5222 (69.7%) | 5267 (70.3%) | 0.012 |
| AST, units/L | 27.5 ± 24.1 | 31.4 ± 49.5 | 0.101 | 27.5 ± 24.1 | 31.1 ± 41.3 | 0.109 |
| Sodium, mEq/L | 137.7 ± 3.2 | 137.8 ± 3.4 | 0.028 | 137.7 ± 3.2 | 137.7 ± 3.4 | 0.005 |

*ACEI* angiotensin-converting enzyme inhibitor, *ARB* angiotensin receptor blocker, *AST* aspartate transaminase, *BNP* B-type natriuretic peptide, *CCB* calcium channel blocker, *COPD* chronic obstructive pulmonary disease, *DPP-4i* dipeptidyl peptidase-4 inhibitor, *eGFR* estimated glomerular filtration rate, *GLP-1* glucagon-like peptide 1, *HbA1C* glycated hemoglobin, *PSM* propensity score matching, *SD* standard deviation, *Std diff* Standardized difference

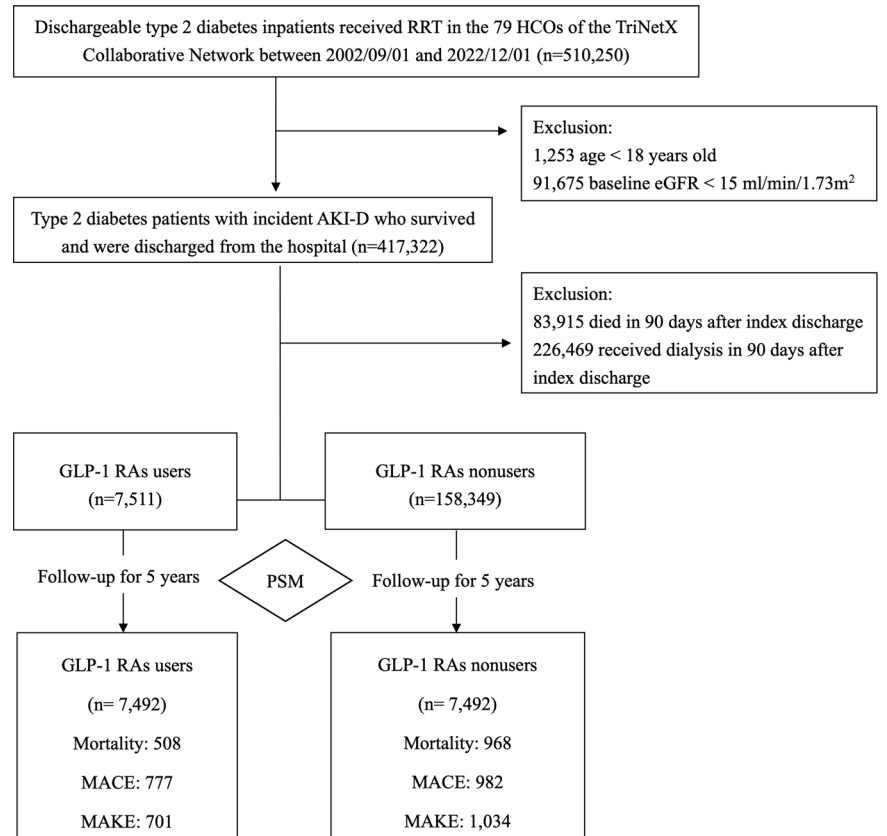

**Fig. 1 | Patient enrollment algorithm.** Flowchart showing the selection of type 2 diabetes patients with AKI-D from the TriNetX Collaborative Network (2002/09/01 – 2022/12/01). A total of 417,322 patients with AKI-D were discharged from the hospital. After exclusions, 165,860 AKD patients were identified, comprising 7,511 GLP-1 RA users and 158,349 nonusers. PSM resulted in 7,492 patients in each group. Five-year follow-up outcomes included mortality, MACE, and MAKE. *AKI-D* dialysis-requiring acute kidney injury, *ESRD* end-stage renal disease, *GLP-1 RAs* glucagon-like peptide 1 receptor agonists, *HCO* healthcare organization, *MACE* major adverse cardiac event, *MAKE* major adverse kidney event, *PSM* propensity score matching, *RRT* renal replacement therapy.

**Table 2 | Incidence of outcomes of interest among the GLP-1 RAs users compared to the control group after PSM**

| Outcome | Patients with outcome GLP-1 RAs group | aHR Control group | (95%CI) |
|---|---|---|---|
| **Primary outcome** | | | |
| Mortality | 6.8% (508/7492) | 12.9% (968/7492) | 0.57 (0.51-0.64) |
| **Secondary outcome** | | | |
| MACE | 14.8% (777/5251) | 18.8% (982/5234) | 0.88 (0.80-0.96) |
| MAKE | 10.8% (701/6513) | 16.0% (1034/6478) | 0.73 (0.66-0.80) |

*aHR* adjusted hazard ratio, *MACE* major adverse cardiac events, *MAKE* major adverse kidney events, *GLP-1 RAs* glucagon-like peptide 1 receptor agonists, *PSM* propensity score matching

Our findings unveil a remarkable survival advantage linked to the use of GLP-1 RAs among the included patients. The consistently higher HbA1C levels in the GLP-1 RAs group throughout the study suggesting the pleiotropic effects of GLP-1RAs additional to glucose lowering effect[21]. GLP-1 RAs are known to stimulate insulin and inhibit glucagon, effectively regulating body weight and glucose levels[13]. Furthermore, the literature indicates that aside from their benefits in weight management and glycemic control, GLP-1 RAs have favorable effects on metabolic profiles, including blood pressure and lipid levels[22]. These favorable effects may contribute to the observed survival benefits in patients with type 2 diabetes and concomitant AKD.

Additionally, patients with diabetes and kidney problems are more prone to sepsis and worse outcomes[23]. Recent research indicates that drugs targeting incretin may help reduce inflammation and blood clotting in sepsis by activating the GLP-1 receptor[24]. In our study, over

50% of patients had sepsis could support findings that GLP-1 RAs could lower the risk of death from sepsis[25]. Moreover, we also found that the survival advantages associated with GLP-1 RAs remain consistent regardless of the concurrent use of other anti-diabetic agents or RAAS blockers. These findings underscore the multifaceted benefits of GLP-1 RAs therapy in patients with type 2 diabetes and AKD.

The cardioprotective effects of GLP-1 RAs in patients diagnosed with type 2 diabetes, alongside associated reductions in cardiovascular risk factors, such as glycated hemoglobin level and BMI are now further elucidated by our findings[17,22,26]. Our study supports the efficacy of GLP-1 RAs in reducing cardiovascular events among individuals with concurrent type 2 diabetes and AKD, highlighting their role in the complex management of these comorbid conditions.

Several cardiovascular outcome trials (CVOTs) have also reported that GLP-1 RAs may be associated with mitigating kidney-related

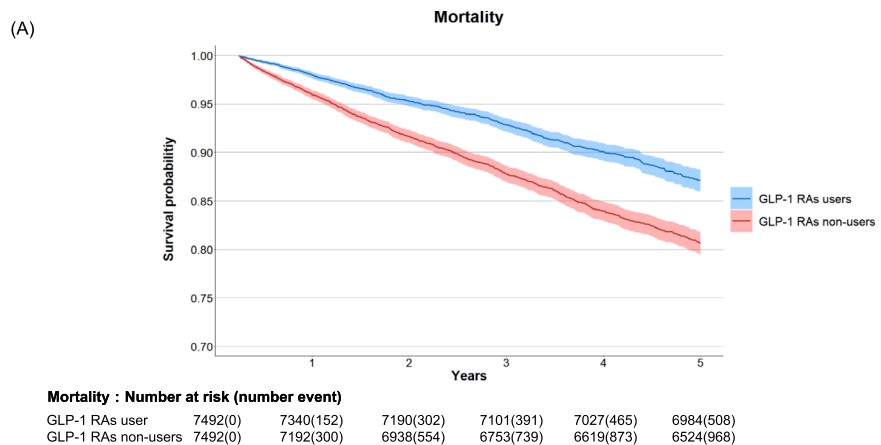

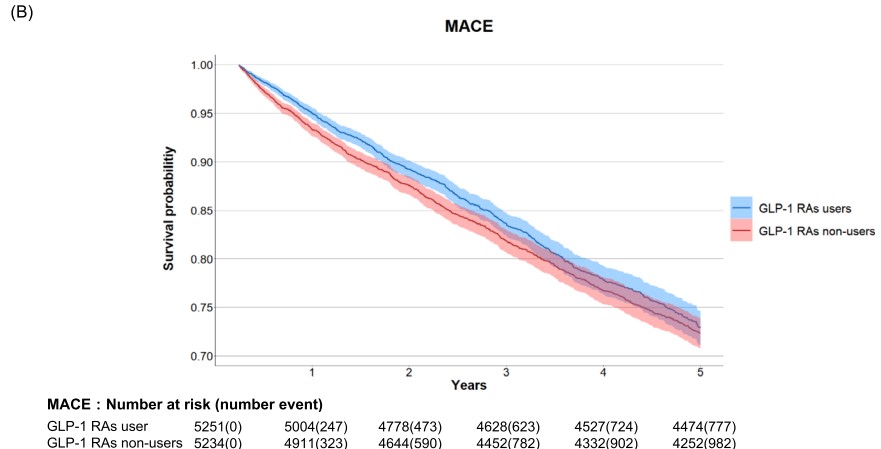

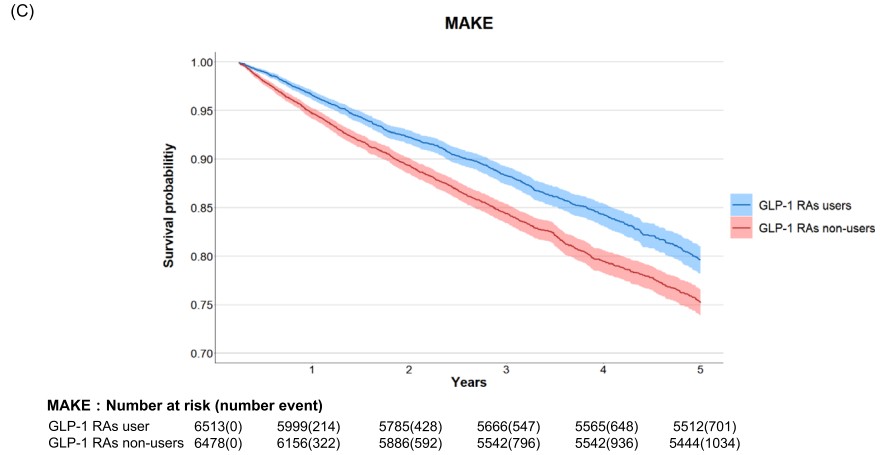

**Fig. 2 | Kaplan-Meier curves showing the long-term outcomes of interest following the use of GLP-1 RAs in a propensity score-matched counterpart.** **A** all-cause mortality (log-rank *P* < 0.001), **B** MACEs (log-rank *P* < 0.001), **C** MAKEs (log-rank *P* < 0.001). The blue line corresponds to GLP-1 RAs users, and the red line represents GLP-1 RAs non-users. Data are presented as mean values with 95% confidence intervals (error bands). The number at risk at different time points is shown below the curves. Source data are provided as a Source Data file. *MACE* major adverse cardiac event, *GLP-1 RAs* glucagon-like peptide 1 receptor agonists, *MAKE* major adverse kidney event.

outcomes, including albuminuria, decline in eGFR and the risk of end-stage kidney disease, independent of their effects on glycemic control[14,15,17,26,27]. The recent FLOW trial, focusing on individuals with type 2 diabetes and chronic kidney disease, suggests that GLP-1 RAs can slow CKD progression[28]. Our study provides additional real-world evidence for the renoprotective effect of GLP-1 RAs in patients with type 2 diabetes and AKD. We further notice that GLP-1 RA users had a

slower eGFR decline, implying a possible effect on kidney function[20,29]. However, our findings don't directly confirm this, highlighting the necessity for further dedicated studies.

The mechanisms behind the renoprotective effects of GLP-1 RAs involve anti-albuminuric, anti-inflammatory, natriuresis, anti-atherogenic, and anti-oxidant effects through protein kinase A signaling, induction of natriuresis by inhibiting sodium-hydrogen exchanger 3, and a reduction

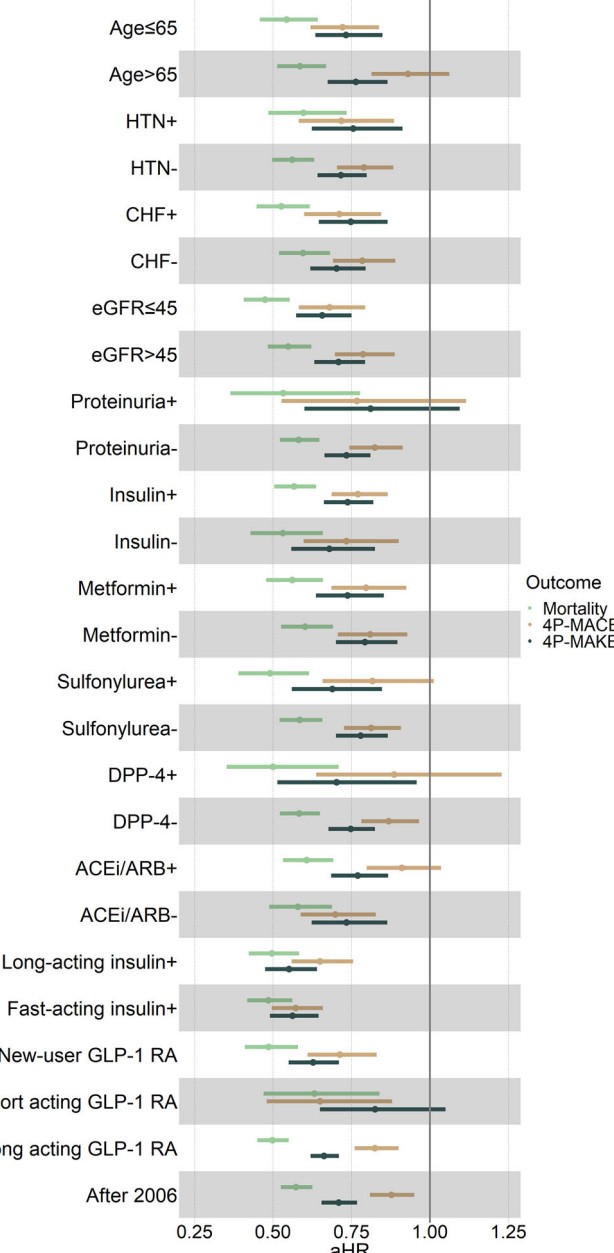

**Fig. 3 | Subgroup analysis.** Forest plots of adjusted hazard ratios (aHRs) for the GLP-1 RAs users (n = 7492) versus non-users (n = 7492) during the AKD period regarding the long-term risks of subgroup analysis for all-cause mortality, MACEs, and MAKEs. The HRs were adjusted for age, sex, and race due to their potential interactions with kidney disease. AHRs (center) and 95% CIs (error bars) are presented. The vertical line indicates an aHR of 1.00; lower limits of 95% CIs with values greater than 1.00 indicate a significantly increased risk. Independent samples were used, with each sample derived from different subjects. Data collection involved independent measurements from each patient. Control groups are defined as non-users of GLP-1 RAs. Source data are provided as a Source Data file. *ACEI* angiotensin-converting enzyme inhibitor, *AHR* adjusted hazard ratio, *AKD* acute kidney disease, *ARB* angiotensin receptor blocker, *DPP-4i* dipeptidyl peptidase-4 inhibitor, *eGFR* estimated glomerular filtration rate, *GLP-1 RAs* glucagon-like peptide 1 receptor agonists, *MACE* major adverse cardiac event, *MAKE* major adverse kidney event. "+" denotes subgroups with additional conditions potentially affecting GLP-1 RAs outcomes, while "-" represents subgroups without these conditions.

in hyperfiltration[27,30,31]. Noteworthily, we observed that long-acting GLP-1 RAs offer more pronounced benefits, potentially due to their sustained pharmacological action, suggesting a dependency on pharmacokinetic profiles for maximal therapeutic impact. In contrast, while short-acting

GLP-1 RAs did not show significant effects on MAKE in our cohort, it is important to note that lixisenatide has shown kidney benefits in the ELIXA trial[32]. This discrepancy underscores the complexity of translating clinical trial results into broader patient populations and suggests that the impact of short-acting GLP-1 RAs on kidney outcomes may vary depending on specific clinical contexts.

Furthermore, the consistent reduction in the risk of MACEs and MAKEs observed with GLP-1 RAs use across patient subgroups underscores the potential of these agents in managing cardiorenal risk. This includes those with hypertension, advanced CKD, and those using insulin or metformin. Our findings highlight the beneficial effects of GLP-1 RAs in improving survival and reducing adverse cardiac and kidney events in individuals with type 2 diabetes and AKD.

Remarkably, the survival benefits of GLP-1RAs, as observed in CVOTs, typically manifest over a more extended period than what our Kaplan Meier curves suggest[15,33]. The rapid divergence observed in mortality, MACE, and MAKE almost immediately after the start of follow-up prompted us to explore alternative mechanisms beyond general metabolic improvements. Early benefits of GLP-1 RAs may derive from acute hemodynamic improvements or direct cellular protective effects in the context of AKI[34−36].

Nikolaidis et al. have demonstrated the salutary cardioprotective effects of GLP-1 RAs following acute myocardial infarction[37]. In our cohort, the combined prevalence of cardiogenic shock and cardiorenal syndrome exceeds one-third, marking a substantial portion of patients at an increased risk of these acute complications. This high prevalence lends further support to the potential immediate impacts of GLP-1 RAs[37], possibly offering protection and aiding in recovery from acute insults. This alignment with our findings and hypotheses underscores the necessity for additional research into the benefits of GLP-1 RAs therapy during dialysis and in acute care settings.

Considering the potential long-term public health challenges posed by AKD[38−40], it is imperative for healthcare providers to incorporate GLP-1 RAs as an integral component of a comprehensive approach aimed at addressing this serious public health concern. However, the remarkably low utilization of GLP-1 RAs among these patients, as shown in this study, underscores an urgent need to heighten awareness and implementation of this therapeutic approach. Our results indicate that the survival benefits and renoprotective effects of GLP-1 RAs remain consistent regardless of the concurrent use of other anti-diabetic agents or RAAS blockers. In addition, the use of GLP-1 RAs has significantly reduced the risk of mortality, MACE, and MAKE in patients requiring metformin or insulin therapy, suggesting the cost-effectiveness of GLP-1 RAs in these high-risk patients[41].

Our study possesses notable strengths but also certain limitations. Firstly, reliance on diagnostic codes for disease classification might have resulted in an underrepresentation of mild cases, potentially leading to ascertainment bias. Furthermore, while MACE included mortality attributed specifically to cardiac causes, the broader definition of mortality within MAKE did not allow for precise attribution to renal causes due to the limitations of diagnostic coding. Despite efforts were made to mitigate misclassification bias and residual confounding through positive and negative controls and the computation of E-values, these risks cannot be completely disregarded. Moreover, while validated outcome definitions and PSM were used to minimize differences in comorbidities and medication usage, the inherent limitations associated with electronic health records remain. Additionally, our analysis could not discern the specific clinical indications for GLP-1RAs initiation during AKD due to the nature of database research, limiting our ability to fully contextualize prescribing decisions. Finally, the lack of detailed reasons for redialysis or mortality also constrains the depth of our outcome analysis,

In summary, our observations suggest that GLP-1 RAs may be associated with a reduction in the risk of mortality, MACEs, and MAKEs in patients with type 2 diabetes following severe AKI, over a median follow-up of 2.3 years. These findings provide insight into the potential benefits of GLP-1 RAs, but they are preliminary and warrant further investigation. Randomized controlled trials are necessary to validate if GLP-1 RAs genuinely enhance the health of these patients and to ensure their safety.

## Methods

### Ethics statement

Data analysis utilizing the TriNetX platform received approval from the Institutional Review Board of Chi-Mei Hospital (No: 11202-002), as well as approval from the institutional review boards of all participating hospitals. Compliance with both the Health Insurance Portability and Accountability Act and General Data Protection Regulation is maintained by the TriNetX platform[42,43]. Since the platform consolidates only de-identified data summaries and counts, TriNetX has been granted an informed consent waiver by the Western Institutional Review Board[42]. Ethical approval for the validation cohort was obtained from the Institutional Review Board of Chang Gung Memorial Hospital (No: 202201889B0). Written informed consent was waived by the IRB due to the retrospective nature of the study. This study was conducted following the principles of the Declaration of Helsinki.

### Data source, study protocol and patient selection

Data for this investigation were accessed through the TriNetX Analytics platform, a global collaborative health research network widely used in numerous prominent epidemiological studies (Supplementary appendix)[43–45]. The dataset used in this study was extensive, encompassing various aspects of patient information including demographic details, diagnoses (according to International Classification of Diseases, Tenth Revision, Clinical Modification [ICD-10-CM] codes), procedures (classified according to the International Classification of Diseases, Tenth Revision, Procedure Coding System [ICD-10-PCS] or Current Procedural Terminology), and medications (coded as per the Veterans Affairs National Formulary). It also covered laboratory tests (organized using Logical Observation Identifiers Names and Codes) and healthcare utilization records from a network of 79 healthcare organizations including hospitals, primary care facilities, and specialist care providers. The extensive dataset incorporated data from both insured and uninsured patients, comprising a participant pool exceeding 250 million individuals. The dataset spans a time period from September 1, 2002, to December 1, 2022. This study adhered to the Strengthening the Reporting of Observational Studies in Epidemiology (STROBE) reporting guideline for cohort studies (Supplementary appendix).

### Prespecified outcomes

The primary outcome was all-cause mortality, and the secondary outcomes were 4-point major adverse cardiac events (MACEs) and major adverse kidney events (MAKEs), 4-point MACEs were defined as stroke (cerebral infarction or hemorrhagic stroke), acute myocardial infarction, cardiac arrest, and mortality, and 4-point MAKEs were defined as redialysis, dialysis dependence, estimated glomerular filtration rate (eGFR) < 15 ml/min/1.73 m² and mortality.

### Covariates

To address disparities between groups, we recorded various factors of the studied population. Essential demographic data included age, sex, and ethnicity, as well as existing comorbidities and medication usage. To mitigate potential discrepancies in baseline characteristics between the two study groups, we comprehensively integrated and algorithmically selected high-dimensional covariates assessed within 1 year

before the index time. The identification of comorbidities was based on ICD-10-CM codes. We also collected potentially influential factors derived from physical examinations, including systolic blood pressure and body mass index (BMI). A comprehensive set of laboratory tests was conducted as part of the analysis, including eGFR, proteinuria, white blood cell, platelet, total cholesterol, glycohemoglobin (HbA1C), aspartate aminotransferase, and sodium.

### Study cohort

We identified 165,860 patients diagnosed with type 2 diabetes and AKD upon admission to the enrolled healthcare facilities during study period (Fig. 1). Patients with AKD were defined as those who were discharged and able to wean from acute kidney injury (AKI) requiring dialysis[46,47]. For all participates, the index date was determined as the 90 days following their hospital discharge. The inclusion criteria were age 18 to 90 years, a confirmed diabetes diagnosis, and ever dialysis during their hospital stay. The patients who had an eGFR <15 ml/min/1.73m² before the index hospitalization and either remained on dialysis, required re-dialysis, or died within 3 months post-discharge were excluded. We categorized patients as GLP-1 RAs users if they had been prescribed a GLP-1 RAs at AKD. The cohort was divided into two groups: the GLP-1 RAs users group ($n = 7,511$), and the GLP-1 RAs non-users group ($n = 158,349$). Propensity score matching (PSM) was performed using 25 variables detailed in the "Covariates" section. All patients were closely tracked for up to 5 years for any occurrence of the outcome of interests. To counteract potential protopathic or ascertainment bias, any instances of primary and secondary outcomes that manifested before the designated index date were disregarded, prompting a repeat of the PSM process.

### Prespecified subgroup analyses

Subgroup analyses were conducted to examine variations in risk related to the desired outcomes among the GLP-1 RAs users. These prespecified analyses considered factors such as age, hypertension, heart failure, eGFR, proteinuria, new users after the index day, enrollee after 2006 and concurrent usage of other medications for glycemic control, and renin-angiotensin-aldosterone system (RAAS) blockers.

Crucially, we also differentiated between short-acting GLP-1 RAs, such as exenatide and lixisenatide, which have immediate postprandial effects; and long-acting GLP-1 RAs, including exenatide once-weekly, liraglutide, albiglutide, dulaglutide, and semaglutide, known for providing more consistent glycemic control[48,49]. This distinction is pivotal to understanding the pharmacokinetic and dynamic influences on the observed clinical outcomes.

### Statistical analysis

Variables were presented in either numerical form (with means and standard deviations) or categorical form (with counts and percentages), depending on the characteristics of the covariates. To mitigate potential confounding variables, we employed PSM, pairing each GLP-1 RAs user with a non-user using a greedy nearest neighbor matching approach integrated within TriNetX. This method accounted for factors such as age, gender, race, comorbidities, medications, and laboratory data, including HbA1c, blood pressure, lipids, and BMI to provide insights into their roles as risk factors in the outcomes of interest. The balance of baseline characteristics in the populations matched by propensity score was evaluated using standardized difference, with a value < 0.1 indicating a minor difference[50]. To minimize multicollinearity issues, preference was given to continuous variables. Additionally, we excluded cases with missing data or those lost to follow-up to maintain data completeness. To mitigate reverse causality effects, the observation period was initiated the day following the index date and continued for a maximum duration of 5 years. We assessed the relationships between the GLP-1 RAs users and the control group concerning

primary and secondary outcomes using the Cox proportional hazards model, which allowed us to calculate adjusted hazard ratios (aHRs)[51]. Dependence among users within matches was accounted for by using robust standard errors. E-values were calculated for pre-specified outcomes to assess the potential influence of unmeasured confounders[52]. The assumption of proportional hazards was evaluated with the generalized Schoenfeld approach on the TriNetX platform, with aHRs recalculated for specific time frames if initial assumptions were not met. E-values, following VanderWeele and Ding's approach, estimated the minimum strength of association needed by unmeasured confounders to fully explain observed associations, addressing potential biases from unmeasured confounding[53,54]. Every analysis included a 95% confidence interval (CI), with statistical significance set at a 2-sided P value < 0.05. The Kaplan-Meier method illustrated survival probabilities. To reinforce the reliability of primary analyses, external validation was performed using data from the Chang Gung Research Database (CGRD)[55]. Furthermore, sensitivity assessments included evaluating cases across different registration periods and Cox proportional analysis with distinct covariates. Analyses focused on specificity, positive outcome controls, and designated negative outcome controls were performed (Supplementary appendix). We used R software (version 3.2.2, Free Software Foundation, Inc, Boston, MA), SAS (version 9.2, SAS Inc., Cary, NC), and Stata/MP (version 16, StataCorp, College Station, TX).

### Reporting summary

Further information on research design is available in the Nature Portfolio Reporting Summary linked to this article.

## Data availability

The datasets used and/or analyzed during this study consist of aggregate data sourced from the TriNetx platform. Due to TriNetx's data sharing policies, we do not have access to individual-level data. Requests for access to the datasets should specify the intended use and will be reviewed and responded to within 2 weeks. Once approved, access to the datasets will be granted within 4 weeks. Access may be subject to limitations based on institutional regulations and data protection policies. For inquiries regarding access, please contact Vin-Cent Wu, the corresponding author, at q91421028@ntu.edu.tw. Source data are provided with this paper.

## Code availability

The R script used for the analyses in this study has been deposited on GitHub and can be accessed at the following link: https://github.com/hcpan1980/KidneyStats/tree/main. The code is publicly available and can be reused without restriction.

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

## Acknowledgements

This study was supported by grants from the Chang Gung Memorial Foundation (CMRPG-5D0111, CMRPG2F0171, CMRPG2F0172, CMRPG2F0173, CMRPG-2G0361, CMRPG-2H0161, CRRPG2H0162, CMRPG-2J0261, CMRPG-2K0091, CORPG2N0141, CORPG2P0231, and CORPG2P0191), the Ministry of Science and Technology, Taiwan (104-2320-B-182A-013, 106-2314-B-182A-064, 107-2314-B-182A-138, 108-2320-B-182A-009-MY3, 108-2314-B-182A-027, 108-2321-B-182-003 and 109-2321-B-182-001) and the Taiwanese Ministry of Health and Welfare (MOHW110-TDU-B-212-124005, MOHW111-TDU-B-212-134005, MOHW112-TDU-B-212-144005). The authors thank the staff of the Community Medicine Research Center of Keelung Chang Gung Memorial Hospital. We also express our sincere gratitude to the administrative support of the Chang Gung Memorial Hospital Clinical Trial Center and all staff of the Taiwan Clinical Trial Consortium (TCTC). VCW had full access to all the data in the study and takes responsibility for the integrity of the data and the accuracy of the data analysis.

## Author contributions

All authors contributed significantly to the design and conduct of the study and acquisition of clinical data. H.C.P. and V.C.W. conceived the study. H.C.P. conducted data analyses and wrote the first draft of the manuscript. F.Y.Y. and T.M.H. supervised all analyses, assumes responsibility for the analyses, and assumes responsibility for data interpretation. H.Y.C., C.Y.S., and T.M.H. conducted external validation. J.Y.C and V.C.W. constructed the dataset and reviewed and commented on the drafts of the manuscript.

## Competing interests

The authors declare no competing interests.
