## [Peer Review File · Nature Communications]

GLP-1 Receptor Agonists' Impact on Cardio-Renal Outcomes and Mortality in T2D with Acute Kidney DiseaseREVIEWER COMMENTS

Reviewer #1 (Remarks to the Author):

The authors examined the effects of GLP-1 RAs in patients with type 2 diabetes and AKD. On the basis of global healthcare data from the TriNetX database, they created comparable groups, each consisting of 7492 patients (GLP-1RA users and nonusers), by propensity score matching. They found that GLP-1 RA users showed lower risks of mortality (43%), MACEs (12%), and MAKEs (27%) after adjusting for covariates. The authors' conclusion is that their results emphasize the importance of individualized treatment strategies incorporating GLP-1 RAs for patients concomitant with AKD (their words, but need to be changed in "for patients with type 2 diabetes and AKD").

COMMENTS

Line 93: The class effect of GLP-1 to reduce MACE has not been demonstrated for all compounds within the class. The American Diabetes Association recommends the use of GLP-1RA with proven cardiovascular benefits in patients with type 2 diabetes and CV disease.

Line 111: The authors quote Ref 20 for describing the TriNetX platform. However, they apparently forgot to acknowledge that Ref. 20 is the brother paper of the present one in which they evaluated the effect of SGLT-2 inhibitors (another class of new anti-hyperglycemic agents) within the same cohort.

They must clearly acknowledge the paper in order to remove the suspicion they want to hide something.

They also need to precise how and why the secondary outcomes (MACEs and MAKEs) in the present study are a little bit different from those of Ref. 20.

Line 156: The authors must also explain why the number of patients in the cohort differs: 230 366 in Ref. 20 and 165 860 in the present paper, despite similar inclusion criteria.

Line 245: I am impressed by the huge reduction (43%) in the relative risk of mortality in GLP-1RA users, which is even greater than the 31% reduction seen in the other paper (Ref. 20).

Was this result a consequence of the different number of patients in the two cohorts? If not, how do the authors explain this?

Line 364: The authors focus on the many limitations of the present study, some of them seems unnecessary. In general, the Discussion appear too long and should be restricted.

Line 393: The conclusion must be less enthusiastic. It must also be stressed that the results are hypothesis-generating and must be validated in specific intervention trials.

Reviewer #2 (Remarks to the Author):

This interesting manuscript by Heng-Chih Pan et al. reports the results of a retrospective observational real-world study in 165,860 patients with type 2 diabetes that were admitted to healthcare facilities with dialysis-requiring acute kidney injury. The investigators examined the long-term effects of GLP-1RA therapy versus no GLP-1RA therapy in this population, employing propensity score matching to create comparable groups, on all-cause mortality (primary outcome), and major adverse cardiovascular and kidney events (secondary outcomes, both defined as a 4-point composite) after a median follow-up of 2.3 years. The results implicate that GLP-1 RAs improves survival and cardiorenal outcomes in this population, after adjusting for covariates, and remained consistent across subgroup and sensitivity analyses.

Although the manuscript and results are of high clinical interest, some critical comments and suggestions may help the authors to improve the manuscript.

MAJOR comments:

1. The observational design of this study requires that both cohorts are identical at the start to ascertain reliable clinical interpretation of the results. This remains somewhat problematic in the current study, despite the rigorous matching efforts by the investigators. The propensity score matching was performed using 25 variables, even including covariates 1 year before the index date. However, important factors that may differ between the cohorts from a clinical perspective were not sufficiently included/taken into account, most notably 1) HbA1c [before, at, and after the index date], 2) eGFR [before and at the index date; preferably an eGFR slope is needed BEFORE the AKI event took place], 3) presumptive causes of AKI. Please clarify why this was not taken into account, amend if possible, and include a detailed discussion how this (likely) affected the outcomes. Please show HbA1c

and eGFR before, at and following the index date (crude numbers; not percentages), using as many timepoints available. A higher HbA1c before the index date may result in increased CV and renal risk (legacy effect), a steeper eGFR slope before the index date may result in an earlier renal event at follow-up; a single timepoint does not cover the full risk of an individual patient or cohort; this issue should be discussed in the limitation section. Why did the authors not match for presumptive cause of AKI, as this may influence recurrence or other outcomes of interest.

2. Please explain the rationale for the covariates that were chosen in the cox regression models. Please add HbA1c, blood pressure, lipids, BMI to the cox-regression models, to gain understanding of the role of these risk factors in explaining the outcomes. If the outcome is explained to large extent by HbA1c, then the drug-specific benefit of GLP-1RA should be questioned. In line, in the light of the sensitivity analysis between GLP-1RA and other active glucose-lowering treatments, please include potential HbA1c differences between groups during the follow-up period.

3. The Kaplan Meier curves showing mortality, MACE and MAKE dissect almost immediately after the start of follow-up (almost within the first month, based on the Figures). The authors state that general metabolic profiles could explain the shown benefit, yet this seems unlikely with these sorts of trajectories. Notably, the benefits of GLP-1RA's in CVOTs take up much more time to occur, and as such alternative/additional explanations should be sought. Could the authors give alternative explanations for the particular population studied? Perhaps hemodynamic benefits, or specific direct effects of GLP-1RA on cardiac/renal cells after being stressed by the cause of AKI (e.g. sepsis) and/or dialysis.

Perhaps include a section on and reference to the salutary effects of GLP-1 directly following acute myocardial infarction/reperfusion on ventricular function and regional functional recovery in the peri-infarct zone, and potential underlying cellular mechanisms at play (e.g. Nikolaidis *Circulation* 2004; PMID 14981009). Perhaps the authors should call for studies that in line with these studies investigate the benefit of GLP-1 during dialysis on outcomes?

4. Patients were categorized as GLP-1RA-users if they had been prescribed a GLP-1RA "at AKD"; please discuss this in more detail, particularly the indications for which the drug was started at this point in time (e.g. in kidney failure due to severe sepsis), given that these agents should be used with caution in this setting based on clinical guidelines and SPC's. Particularly those GLP-1RA's with an exendin backbone (all short-acting GLP-1RA's) which

are cleared by the kidneys, and in the light of the early years after GLP-1RA's were introduced (2005-2013) with case reports of acute kidney failure as a result of GLP-1RA use were published (e.g. Filippatos World J Diabetes 2013; PMID 24147203).

MINOR comments:

1. The dataset spans a time period from September 2002 onward; what was the rationale for the 2002 timepoint in this study, as GLP-1RA's were not available until 2004/2005.
2. Title (Page 1): Please include the outcomes investigated (ie. Cardio-Renal Outcomes and Morality), and that this is a Retrospective Observational Cohort study from the TriNetX Collaborative Network.
3. Abstract: Please include the main causes of AKI in this study (e.g. 55.2% due to sepsis, 34.2% due to cardiorenal syndrome).
4. Methods/Discussion: Please indicate which GLP-1RA's were used in this study, and which were classified as short-acting and long-acting compounds. In the discussion, please elaborate specifically how the pharmacokinetic profiles of these compounds may result in differences in pharmacodynamic effects in type 2 diabetes patients (i.e. tachyphylaxis). Please add that there was a renal benefit of the short-acting GLP-1RA lixisenatide in ELIXA (Musket, Lancet DE 2018, PMID: 30292589)
5. Methods (Page 9; prespecified outcomes): Was mortality in the MAKE and MACE defined as all-cause mortality, or death due to renal and cardiovascular disease respectively?
6. Methods (Page 10, Study Cohort, Line 158): Please rephrase the definition of the index date; it seems it was 90 days following hospital discharge; this marks an important factor of the study.
7. Results (Page 17): Do the authors have an explanation why the negative outcomes all seem to be numerically higher in the GLP-1RA group, with some even almost reaching significance? Note that Crohn's disease is spelled incorrectly in the text and supplementary table.
8. Discussion (Page 21, line 338): Please show data that indicate that GLP-1RA's reduce hyperfiltration, as this -as far as I'm aware- has not been shown in eGFR trajectories of large outcome trials (while it is seen in those studying SGLT2 inhibitors), and specific mechanistic studies in humans did not find a beneficial effect of these drugs on measured (intra)renal hemodynamics.

Reviewer #1 (Remarks to the Author):

The authors examined the effects of GLP-1 RAs in patients with type 2 diabetes and AKD. On the basis of global healthcare data from the TriNetX database, they created comparable groups, each consisting of 7492 patients (GLP-1RA users and nonusers), by propensity score matching. They found that GLP-1 RA users showed lower risks of mortality (43%), MACEs (12%), and MAKEs (27%) after adjusting for covariates. The authors' conclusion is that their results emphasize the importance of individualized treatment strategies incorporating GLP-1 RAs for patients concomitant with AKD (their words, but need to be changed in "for patients with type 2 diabetes and AKD").

Response: thank you for your comments, we have revised the words.

COMMENTS

Line 93: The class effect of GLP-1 to reduce MACE has not been demonstrated for all compounds within the class. The American Diabetes Association recommends the use of GLP-1RA with proven cardiovascular benefits in patients with type 2 diabetes and CV disease.

Response: Thanks for your comment. We have revised and emphasized the associated sentences in the "Introduction" section (P.6-P.7).

"While the studies supporting these benefits were not initially focused on kidney health and included patients with low kidney risk,

they suggest a link between GLP-1 RAs treatment and kidney protection, especially for type 2 diabetes patients with CKD[1]. “

[Reference]

1. Shaman, A.M., *et al.* Effect of the glucagon-like peptide-1 receptor agonists semaglutide and liraglutide on kidney outcomes in patients with type 2 diabetes: pooled analysis of SUSTAIN 6 and LEADER. *Circulation* **145**, 575-585 (2022).

Line 111: The authors quote Ref 20 for describing the TriNetX platform. However, they apparently forgot to acknowledge that Ref. 20 is the brother paper of the present one in which they evaluated the effect of SGLT-2 inhibitors (another class of new anti-hyperglycemic agents) within the same cohort.

They must clearly acknowledge the paper in order to remove the suspicion they want to hide something.

They also need to precise how and why the secondary outcomes (MACEs and MAKEs) in the present study are a little bit different from those of Ref. 20.

Response: Thanks for your comment. We have addressed the concerns raised by adding a relevant section about the methodological divergence and outcome analysis between the current study and our previous SGLT-2 inhibitors study in the supplementary file (Suppl. P.10). In this study, adjustments have been made to the definitions and analyses of secondary outcomes compared to our prior investigation utilizing the TriNetX platform to evaluate SGLT-2 inhibitors (as referenced in Ref. 20). Specifically, we have included "eGFR < 15 ml/min/1.73m²" as part of the criteria for MAKEs. Consequently, AKD patients with baseline kidney

function less than 15 ml/min/1.73m² have been excluded from the analysis. Moreover, in contrast to the previous study where cardiogenic shock was included in the criteria for MACEs, this investigation has replaced it with cardiogenic arrest to better align with the outcomes observed in previous studies of GLP-1 receptor agonists[1-3]. These adjustments aim to improve the specificity and clinical relevance of our findings regarding the effects of GLP-1 RAs.

We have revised the patient enrollment algorithm in the Figure 1 for clarification and also revised the citation regarding the description of the TriNetX platform to accurately acknowledge its source[4]. This acknowledgment is essential to ensure transparency and to alleviate any concerns regarding the intention to conceal information.

Revised Figure 1. Patient enrollment algorithm.

Abbreviations: AKI-D, dialysis-requiring acute kidney injury; ESRD, end-stage renal disease; GLP-1 RAs; glucagon-like peptide 1 receptor agonists; HCO, healthcare organization; MACE, major adverse cardiac event; MAKE, major adverse kidney event; PSM, propensity score matching; RRT, renal replacement therapy.

[Reference]

1. Marso, S.P., *et al.* Semaglutide and cardiovascular outcomes in patients with type 2 diabetes. *New England Journal of Medicine* **375**, 1834-1844 (2016).
2. Marso, S.P., *et al.* Liraglutide and cardiovascular outcomes in type 2 diabetes. *New England Journal of Medicine* **375**, 311-322 (2016).
3. Holman, R.R., *et al.* Effects of once-weekly exenatide on

cardiovascular outcomes in type 2 diabetes. *New England Journal of Medicine* **377**, 1228-1239 (2017).

4. Topaloglu, U. & Palchuk, M.B. Using a Federated Network of Real-World Data to Optimize Clinical Trials Operations. *JCO Clin Cancer Inform* **2**, 1-10 (2018).

Line 156: The authors must also explain why the number of patients in the cohort differs: 230 366 in Ref. 20 and 165 860 in the present paper, despite similar inclusion criteria.

Response: Thanks for your comment regarding the different patient numbers between our current study and Ref. 20, the variation can be primarily attributed to the inclusion of a four-point MAKE criterion, which introduces “eGFR < 15 ml/min/1.73m²” as a qualifying event. The inclusion of this additional criterion resulted in the exclusion of patients who would otherwise qualify for the cohort under the three-point MAKE definition used in Ref. 20. This exclusion is significant as patients with an eGFR < 15 are more likely to be undergoing dialysis due to CKD progression, rather than as a direct result of an AKI event.

We believe that the use of a four-point MAKE, including “eGFR < 15 ml/min/1.73m²” offers a more precise tool for evaluating the impact of GLP-1 RAs on kidney function, particularly in the context of AKI. This distinction is crucial because patients with advanced CKD are typically managed differently, and their inclusion could potentially confound outcomes related to AKI events. Removing patients with eGFR < 15 ml/min/1.73m² can allow for a more focused study on AKI, as many of these patients requiring RRT are due to CKD progression rather than AKI. By focusing on patients less likely to be affected by CKD progression, we

can more accurately assess the relationship between GLP-1 RA use and kidney-related outcomes post-AKI.

Therefore, while the cohort in the present study is smaller, the design offers a more targeted approach to understand the potential benefits and risks of GLP-1 RAs in a population that is more homogeneously impacted by AKI. This nuanced approach enhances the study's validity by ensuring that the observed outcomes are more likely attributable to the effects of the intervention rather than the underlying chronic disease progression.

To further clarify this point, we have revised the patient enrollment algorithm in the Figure 1 and exclusion criteria in the “Method” section (P.10). ensuring transparency and consistency in our approach.

Revised Figure 1. Patient enrollment algorithm.

Abbreviations: AKI-D, dialysis-requiring acute kidney injury; ESRD, end-stage renal disease; GLP-1 RAs; glucagon-like peptide 1 receptor agonists; HCO, healthcare organization; MACE, major adverse cardiac event; MAKE, major adverse kidney event; PSM, propensity score matching; RRT, renal replacement therapy.

Line 245: I am impressed by the huge reduction (43%) in the relative risk of mortality in GLP-1RA users, which is even greater than the 31% reduction seen in the other paper (Ref. 20). Was this result a consequence of the different number of patients in the two cohorts? If not, how do the authors explain this?

Response: Thanks for your comment regarding to the pronounced reduction in the relative risk of mortality observed in our study with GLP-1 RA users, particularly when contrasted with the results from Ref. 20. The divergence in outcomes may, in part, be elucidated by the differences in the prevalence of baseline comorbidities between the two groups. The cohort data reveal disparities in key comorbidities such as chronic kidney disease, heart failure, and ischemic heart diseases. For example, chronic kidney disease was present in 32.4% of the GLP-1 RAs group versus 34.0% in the SGLT-2 inhibitors group from Ref. 20. Likewise, congestive heart failure and ischemic heart diseases were reported in 27.8% and 38.1% of the GLP-1 RAs group, respectively, compared to 51.1% and 55.8% in the SGLT-2 inhibitors group. Moreover, GLP-1 RAs users were younger on average, with a mean age of 59.0 years compared to 63.8 years in the SGLT-2 inhibitors users. This age difference suggests that the GLP-1 RAs users might have been at an earlier stage of disease progression.

Patients with fewer comorbidities and at a younger age are likely to experience a more favorable prognosis, which may be further amplified by the therapeutic effects of GLP-1 RAs. Additionally, we have included “eGFR < 15 ml/min/1.73m²” as part of the MAKE criteria in the current study. The inclusion of this additional criterion resulted in the exclusion of patients who would otherwise qualify for the cohort under the three-point MAKE definition used in Ref. 20. This exclusion is significant as patients with an eGFR < 15 ml/min/1.73m² are more likely to be undergoing dialysis due to CKD progression, rather than as a direct result of an AKI event. The exclusion of patients with eGFR < 15 ml/min/1.73m², a group more prone to complications and possibly less responsive to treatment due to advanced chronic kidney disease, likely

contributed to a clearer demonstration of the benefits associated with GLP-1 RAs use. The cardioprotective actions of GLP-1 RAs, which include glycemic control, blood pressure reduction, and anti-inflammatory properties, could be more effectively manifested in such a population, resulting in the more pronounced mortality risk reduction seen in our study. We have added the comparisons in supplementary results (Suppl p.10).

Line 364: The authors focus on the many limitations of the present study, some of them seems unnecessary. In general, the Discussion appear too long and should be restricted.

Response: Thanks for your comment. We have revised and shortened the limitation part in the “Discussion” section (P.25-26).

Line 393: The conclusion must be less enthusiastic. It must also be stressed that the results are hypothesis-generating and must be validated in specific intervention trials.

Response: Thank you for your valuable suggestion. We have revised the tone in line with your request in the "Discussion" section but also our conclusion (P. 26), ensuring a more neutral expression regarding limitations.

“These findings provide insight into the potential benefits of GLP-1 RAs, but they are preliminary and warrant further investigation. Randomized

controlled trials are necessary to validate if GLP-1 RAs genuinely enhance the health of these patients and to ensure their safety.”

Reviewer #2 (Remarks to the Author):

This interesting manuscript by Heng-Chih Pan et al. reports the results of a retrospective observational real-world study in 165,860 patients with type 2 diabetes that were admitted to healthcare facilities with dialysis-requiring acute kidney injury. The investigators examined the long-term effects of GLP-1RA therapy versus no GLP-1RA therapy in this population, employing propensity score matching to create comparable groups, on all-cause mortality (primary outcome), and major adverse cardiovascular and kidney events (secondary outcomes, both defined as a 4-point composite) after a median follow-up of 2.3 years. The results implicate that GLP-1 RAs improves survival and cardiorenal outcomes in this population, after adjusting for covariates, and remained consistent across subgroup and sensitivity analyses.

Although the manuscript and results are of high clinical interest, some critical comments and suggestions may help the authors to improve the manuscript.

MAJOR comments:

1. The observational design of this study requires that both cohorts are identical at the start to ascertain reliable clinical interpretation of the results. This remains somewhat problematic in the current study, despite the rigorous matching efforts by the investigators. The propensity score matching was performed using 25 variables, even including covariates 1

year before the index date. However, important factors that may differ between the cohorts from a clinical perspective were not sufficiently included/taken into account, most notably 1) HbA1c [before, at, and after the index date], 2) eGFR [before and at the index date; preferably an eGFR slope is needed BEFORE the AKI event took place], 3) presumptive causes of AKI. Please clarify why this was not taken into account, amend if possible, and include a detailed discussion how this (likely) affected the outcomes. Please show HbA1c and eGFR before, at and following the index date (crude numbers; not percentages), using as many timepoints available. A higher HbA1c before the index date may result in increased CV and renal risk (legacy effect), a steeper eGFR slope before the index date may result in an earlier renal event at follow-up; a single timepoint does not cover the full risk of an individual patient or cohort; this issue should be discussed in the limitation section. Why did the authors not match for presumptive cause of AKI, as this may influence recurrence or other outcomes of interest.

Response: We appreciate your insightful comments and have conducted a thorough review of our data, focusing particularly on the HbA1c and eGFR metrics as well as presumptive causes of AKI.

Upon further analysis, our initial observations did not show significant statistical differences between the two groups in the study period (HbA1c p-value = 0.2556, eGFR p-value = 0.818). However, our repeated evaluations revealed that, throughout the AKD, HbA1C levels were consistently higher in the GLP-1 RAs group compared to the control group, indicating that the reduced risk of adverse outcomes associated with GLP-1 RAs is not mainly attributable to the “legacy effect.”

Additionally, the sustained difference in HbA1C levels between the GLP-1 RAs group and the control group prompts us to consider the role of pleiotropic effect additional to glycemic control in the cardiovascular risk reduction observed with GLP-1 RAs therapy, as suggested in the literature[1].

In terms of kidney function, while the initial eGFR readings were significantly lower in the GLP-1 RAs group, this difference diminished by the D60-90 interval after index date, suggesting a potential stabilization or improvement in kidney function with GLP-1 RAs treatment. The hypothesis regarding reductions in glomerular hyperfiltration as a renoprotective mechanism aligns with established theories, although it did not directly emerge from our findings[2-3].

We acknowledge the importance of considering the influence of presumptive AKI causes on the study's outcomes. In response to your concerns, we have conducted a series of sensitivity analyses to address this issue in the supplement file (Table S7). Our results confirmed the robustness of our primary results. This crucial step reinforces the validity of our findings, affirming that the observed benefits of GLP-1 RAs were not confounded by the initial reasons for renal injury.

We have revised the “Result” and “Discussion” section in our manuscript (P.15, P.17, P. 20, and P. 22) and added two new suppl figures (Figure S1 and S2) to provide an in-depth exploration of the various factors impacting our cohort's results.

Suppl. Figure 1. Comparative HbA1C mean levels at baseline, D0-30, and D60-90 for GLP-1 RAs users and the control group.

Abbreviations: GLP-1 RAs; glucagon-like peptide-1 receptor agonists; HbA1C, glycated hemoglobin

Suppl. Figure 2. Comparative eGFR mean levels at baseline, D0-30, and D60-90 for GLP-1 RAs users and the control group.

Abbreviations: eGFR, estimated glomerular filtration rate; GLP-1 RAs; glucagon-like peptide-1 receptor agonists.

[Reference]

1. Scheen, A.J. Cardiovascular protection significantly depends on HbA1c improvement with GLP-1RAs but not with SGLT2 in type 2 diabetes: A narrative review. *Diabetes Metab* **50**, 101508 (2024).
2. Tonneijck, L., *et al.* Glomerular Hyperfiltration in Diabetes: Mechanisms, Clinical Significance, and Treatment. *J Am Soc Nephrol* **28**, 1023-1039 (2017).
3. Muskiet, M.H.A., *et al.* GLP-1 and the kidney: from physiology to pharmacology and outcomes in diabetes. *Nat Rev Nephrol* **13**, 605-628 (2017).

2. Please explain the rationale for the covariates that were chosen in the cox regression models. Please add HbA1c, blood pressure, lipids, BMI to the cox-regression models, to gain understanding of the role of these risk factors in explaining the outcomes. If the outcome is explained to large extent by HbA1c, then the drug-specific benefit of GLP-1RA should be questioned. In line, in the light of the sensitivity analysis between GLP-1RA and other active glucose-lowering treatments, please include potential HbA1c differences between groups during the follow-up period.

Response: Thank you for your inquiry regarding the rationale behind our selection of covariates for the Cox regression models. The selected covariates were as outlined in Table 1, which are well-established risk factors for cardiovascular and renal outcomes in diabetes, and their inclusion is intended to provide a comprehensive assessment of these risk factors on the outcomes.

We appreciate the astuteness of your query regarding the inclusion of covariates such as HbA1c, blood pressure, lipids, and BMI in our Cox regression models. These variables, recognized for their strong association with cardiovascular and renal outcomes in patients with diabetes, were indeed incorporated to mitigate potential confounding effects (table 1). By including these factors, we aimed to refine our models and underscore the independent association between GLP-1 RAs use and the targeted outcomes. Your insightful inquiry has reinforced the robustness of our analysis (please see table 1)

In response to the second aspect of your query, we recognize the significance of assessing HbA1c levels between groups treated with GLP-

1 RAs and other active glucose-lowering therapies during the follow-up period. Our detailed analysis indicates that HbA1c levels were consistently higher in the GLP-1 receptor agonist group compared to the group receiving other glucose-lowering treatments. This observation suggests that the decreased risk of adverse outcomes in the GLP-1 RAs group may not be directly related to improved glycemic control. Instead, it points towards the broader multifaceted benefits of GLP-1 receptor agonists in addition to glycemic control.

We have revised the “Method” and “Result” section in our manuscript (P.12 and P.18) and added a new suppl figure (Figure S3) to address your comments.

Suppl. Figure 3. Comparative HbA1C mean levels at baseline, D0-30, and D60-90 for GLP-1 RAs users and other second-line antihyperglycemic treatments users.

Abbreviations: GLP-1 RAs; glucagon-like peptide-1 receptor agonists; HbA1C, glycated hemoglobin

3. The Kaplan Meier curves showing mortality, MACE and MAKE dissect almost immediately after the start of follow-up (almost within the first month, based on the Figures). The authors state that general metabolic profiles could explain the shown benefit, yet this seems unlikely with these sorts of trajectories. Notably, the benefits of GLP-1RA's in CVOTs take up much more time to occur, and as such alternative/additional explanations should be sought. Could the authors give alternative explanations for the particular population studied? Perhaps hemodynamic benefits, or specific direct effects of GLP-1RA on cardiac/renal cells after being stressed by the cause of AKI (e.g. sepsis) and/or dialysis. Perhaps include a section on and reference to the salutary effects of GLP-1 directly following acute myocardial infarction/reperfusion on ventricular function and regional functional recovery in the peri-infarct zone, and potential underlying cellular mechanisms at play (e.g. Nikolaidis Circulation 2004; PMID 14981009). Perhaps the authors should call for studies that in line with these studies investigate the benefit of GLP-1 during dialysis on outcomes?

Response: We thank you for your insightful observations regarding the rapid divergence observed in the Kaplan Meier curves for mortality, MACE, and MAKE, and the suggestion to explore alternative or additional explanations for the benefits of GLP-1 RAs therapy in our study population.

Thank you to point out that the survival benefits of GLP-1 RAs, as

observed in CVOTs, typically manifest over a more extended period than what our Kaplan Meier curves suggest. This discrepancy prompts a deeper investigation into the potential mechanisms that might contribute to the early benefits observed.

1. Acute hemodynamic benefits: GLP-1 RAs are known to exert acute cardiovascular effects, including improvements in blood pressure and endothelial function, which could have immediate benefits in patients recovering from AKI, particularly those with underlying cardiovascular conditions. These hemodynamic changes might offer an early protective effect, even before the long-term metabolic improvements take hold [1-2].

2. Direct cellular effects: There is emerging evidence suggesting that GLP-1 RAs may have direct cardioprotective effects at the cellular level. For example, studies have indicated that GLP-1 RAs can reduce oxidative stress and inflammation, enhance autophagy, and improve cellular energy metabolism, which could be particularly beneficial in the acute setting of AKI, where oxidative stress and inflammation are prevalent [3].

3. Post-acute myocardial infarction/reperfusion injury: We acknowledge the relevance of studies like that by Nikolaidis et al., *Circulation* 2004, which demonstrated the salutary effects of GLP-1 on ventricular function and regional functional recovery in the peri-infarct zone following acute myocardial infarction/reperfusion injury. Importantly, in our cohort, the combined prevalence of cardiogenic shock and cardiorenal syndrome exceeds one-third, further underscoring the potential for GLP-1 RAs to offer immediate benefits in stabilizing cardiac function and facilitating recovery in acute settings. This high prevalence suggests that our findings of early divergence in outcomes may, in part, be attributable to the immediate hemodynamic and cellular effects of GLP-1 RAs in patients with significant cardiorenal compromise[4].

Additionally, in the “Discussion” section of our manuscript, we have elaborated on the significant potential of GLP-1 RAs to improve outcomes related to sepsis as the leading cause of AKI from our cohort. This is particularly relevant given the increased susceptibility of patients with diabetes and kidney impairment to sepsis and the associated poor outcomes[5]. Our observations, along with emerging evidence, suggest that incretin-based therapy could mitigate excessive inflammation and microvascular thrombosis in sepsis through the activation of the GLP-1 receptor, aligning with the findings by Chen et al. that GLP-1 RAs might offer protective benefits against sepsis-related mortality [6-7]. This addition underscores the multifaceted potential of GLP-1 RAs therapy beyond the traditional metabolic benefits, potentially offering a crucial advantage in managing sepsis outcomes in this vulnerable population.

Given the rapid effects observed and the potential mechanisms discussed, we propose that future studies should explicitly investigate the short-term benefits of GLP-1 RAs therapy in populations at risk of or recovering from AKI, including those undergoing dialysis. Such research would be invaluable in delineating the acute versus chronic benefits of GLP-1 RAs and understanding the optimal timing and patient population for these therapies.

We have revised our manuscript to include these discussions, providing a more comprehensive view of the potential mechanisms through which GLP-1 RAs might exert their benefits in the studied population (P.20-21, P.23-24). We also echo the call for further studies aimed at investigating these mechanisms in detail, particularly in the context of acute cardiac and renal events.

[Reference]

1. Goodwill, A.G., *et al.* Cardiovascular and hemodynamic effects of

- glucagon-like peptide-1. *Rev Endocr Metab Disord* **15**, 209-217 (2014).
2. Zhou, X., *et al.* Acute hemodynamic and renal effects of glucagon-like peptide 1 analog and dipeptidyl peptidase-4 inhibitor in rats. *Cardiovasc Diabetol* **14**, 29 (2015).
 3. Pandey, S., Mangmool, S. & Parichatikanond, W. Multifaceted Roles of GLP-1 and Its Analogs: A Review on Molecular Mechanisms with a Cardiotherapeutic Perspective. *Pharmaceuticals (Basel)* **16**(2023).
 4. Nikolaidis, L.A., *et al.* Effects of glucagon-like peptide-1 in patients with acute myocardial infarction and left ventricular dysfunction after successful reperfusion. *Circulation* **109**, 962-965 (2004).
 5. Schuetz, P., Castro, P. & Shapiro, N.I. Diabetes and sepsis: preclinical findings and clinical relevance. *Diabetes Care* **34**, 771-778 (2011).
 6. Shah, F.A., *et al.* Therapeutic Effects of Endogenous Incretin Hormones and Exogenous Incretin-Based Medications in Sepsis. *J Clin Endocrinol Metab* **104**, 5274-5284 (2019).
 7. Chen, J.-J., *et al.* Association of glucagon-like peptide-1 receptor agonist vs dipeptidyl peptidase-4 inhibitor use with mortality among patients with type 2 diabetes and advanced chronic kidney disease. *JAMA Network Open* **5**, e221169-e221169 (2022).

4. Patients were categorized as GLP-1RA-users if they had been prescribed a GLP-1RA “at AKD”; please discuss this in more detail, particularly the indications for which the drug was started at this point in time (e.g. in kidney failure due to severe sepsis), given that these agents should be used with caution in this setting based on clinical guidelines and SPC’s. Particularly those GLP-1RA’s with an exendin backbone (all short-acting GLP-1RA’s) which are cleared by the kidneys, and in the light of the early years after GLP-1RA’s were introduced (2005-2013) with case reports of acute kidney failure as a result of GLP-1RA use were published (e.g. Filippatos *World J Diabetes* 2013; PMID 24147203).

Response: We thank for your insightful observations on GLP-1 RAs prescriptions during AKD and the use of exendin-based GLP-1 RAs in patients with kidney impairment. Conducting our study with a retrospective healthcare database posed inherent limitations, notably the lack of detailed clinical notes to precisely determine the indications for GLP-1 RAs initiation during AKD. This common challenge in database research is particularly pertinent given the cautious clinical approach recommended for GLP-1 RAs use in patients with kidney impairment.

The period of our study, from 2002 to 2022, was marked by evolving clinical practices regarding GLP-1 RAs use in kidney impairment. We recognize that our inability to ascertain the specific reasons for GLP-1 RAs therapy initiation precludes a definitive analysis of their indications in our AKD cohort. Most of them (49.6%) were ever users before index hospital discharge. In response, we've conducted sensitivity analyses focusing on patients prescribed exendin-based GLP-1 RAs, acknowledging their contraindication in severe kidney dysfunction (eGFR < 30 ml/min/1.73m² as per the current clinical recommendation) [1]. Although these analyses affirmed the consistency with the primary analysis, the limited number of patients in this subgroup resulted in findings that were not statistically significant, as anticipated.

We appreciate the opportunity to delve deeper into the nuances of our data and believe that these additional analyses enrich the overall findings of our study by clarifying the potential risks and benefits of GLP-1 RAs therapy, especially in the presence of renal challenges. In the revised manuscript, we've expanded the "Result" section, limitation part in the "Discussion" section, and Supplementary file (P.17, P. 25-26, Suppl P.9, table S8 and S9) to highlight the lack of detailed prescribing rationales and the potential for confounding by indication.

Table S8. Incidence of outcomes of interest among the GLP-1 RAs users compared to the control group after propensity score matching, in patients treated with Exenatide and an eGFR \geq 30 ml/min/1.73m²

Outcome	Patients with outcome		aHR (95%CI)
	GLP-1 RAs group	Control group	
Primary outcome			
Mortality	16.1% (74/461)	14.5% (67/461)	0.93 (0.67-1.30)
Secondary outcome			
MACE	18.7% (57/305)	19.9% (58/292)	0.77 (0.54-1.12)
MAKE	19.3% (75/388)	18.1% (70/387)	0.90 (0.65-1.24)

Abbreviations: aHR, adjusted hazard ratio; MACE, MAKE, GLP-1 RAs; glucagon-like peptide 1 receptor agonists

Table S9. Incidence of outcomes of interest among the GLP-1 RAs users compared to the control group after propensity score matching, in patients treated with Exenatide and an eGFR < 30 ml/min/1.73m²

Outcome	Patients with outcome		aHR (95%CI)
	GLP-1 RAs group	Control group	
Primary outcome			
Mortality	20.3% (16/79)	26.6% (21/79)	0.65 (0.34-1.25)
Secondary outcome			
MACE	28.6% (12/42)	29.1% (16/55)	0.76 (0.40-1.81)
MAKE	34.0% (18/53)	33.3% (14/42)	0.78 (0.39-1.58)

Abbreviations: aHR, adjusted hazard ratio; MACE, MAKE, GLP-1 RAs; glucagon-like peptide 1 receptor agonists

[Reference]

1. Muskiet, M.H.A., *et al.* GLP-1 and the kidney: from physiology to pharmacology and outcomes in diabetes. *Nat Rev Nephrol* **13**, 605-628 (2017).

MINOR comments:

1. The dataset spans a time period from September 2002 onward; what was the rationale for the 2002 timepoint in this study, as GLP-1RA's were not available until 2004/2005.

Response: Thank you for your insightful observation regarding the starting point of our dataset in September 2002, especially in light of the fact that GLP-1 RAs were not introduced until April 2005[1]. Your critique is both valid and appreciated, as it highlights a critical aspect of our study's design and its alignment with the timeline of GLP-1 RA availability.

In response to your concern, we agree that initiating the dataset analysis from a point prior to the availability of GLP-1 RAs could potentially impact the interpretation of our findings. To address this and ensure the robustness of our conclusions, we have conducted a sensitivity analysis by adjusting the start point of our dataset from Jan 2006 to Dec 2022. This adjustment allows for a period of GLP-1 RAs establishment in clinical practice, thereby ensuring that the patients included in our study could have had access to these medications.

The results from this sensitivity analysis were consistent with our original findings, indicating that the observed associations between GLP-1 RAs use and the outcomes of interest are not artifacts of the dataset's initial time frame. We believe this strengthens the validity of our conclusions and appreciate the opportunity to clarify this aspect of our methodology.

We have included the details of this sensitivity analysis in the revised Figure 3 and supplement appendix (Suppl. P. 9), ensuring that readers are aware of both the rationale behind the original dataset timeframe and the steps taken to validate our findings against potential temporal biases.

We are grateful for your meticulous review and valuable feedback, which have significantly contributed to enhancing the quality and rigor of our work.

Revised Figure 3. Subgroup analysis. Forest plots of adjusted hazard ratios for the GLP-1 RAs users versus non-users during the AKD period regarding the long-term risks of sensitivity analysis for all-cause mortality, MACEs, and MAKEs. The hazard ratios were adjusted for age, sex, and race due to their potential interactions with kidney disease. Adjusted HRs and 95% CIs (error bars) are presented. The vertical line indicates an HR of 1.00; lower limits of 95% CIs with values greater than 1.00 indicate a significantly increased risk.

Abbreviations: ACEI, angiotensin-converting enzyme inhibitor; AKD, acute kidney disease; ARB, angiotensin receptor blocker; DPP-4i, dipeptidyl peptidase-4 inhibitor; eGFR, estimated glomerular filtration rate; GLP-1 RAs; glucagon-like peptide 1 receptor agonists; HR, hazard ratio; MACE, major adverse cardiac event; MAKE, major adverse kidney event

"+" denotes subgroups with additional conditions potentially affecting

GLP-1 RAs outcomes, while "-" represents subgroups without these conditions.

[Reference]

1. Sheahan, K.H., Wahlberg, E.A. & Gilbert, M.P. An overview of GLP-1 agonists and recent cardiovascular outcomes trials. *Postgrad Med J* **96**, 156-161 (2020).

2. Title (Page 1): Please include the outcomes investigated (ie. Cardio-Renal Outcomes and Morality), and that this is a Retrospective Observational Cohort study from the TriNetX Collaborative Network.

Response: Thank you for your valuable suggestion, we have revised the title accordingly (P.1). However this article is also verified by local multicenter cohorts. Therefore, we kindly request your understanding as we revised the title to be "Investigating the Impact of Glucagon-Like Peptide-1 Receptor Agonists on Cardio-Renal Outcomes and Mortality in Type 2 Diabetes with Acute Kidney Disease: A Retrospective Observational Cohort Study".

3. Abstract: Please include the main causes of AKI in this study (e.g. 55.2% due to sepsis, 34.2% due to cardiorenal syndrome).

Response: Thank you for your valuable suggestion, we have revised the abstract accordingly (P.4).

4. Methods/Discussion: Please indicate which GLP-1RA's were used in this study, and which were classified as short-acting and long-acting compounds. In the discussion, please elaborate specifically how the pharmacokinetic profiles of these compounds may result in differences in pharmacodynamic effects in type 2 diabetes patients (i.e. tachyphylaxis). Please add that there was a renal benefit of the short-acting GLP-1RA lixisenatide in ELIXA (Muskiel, Lancet DE 2018, PMID: 30292589)

Response: Thanks for your suggestion. In our study, we included a range of GLP-1 RAs, both short-acting and long-acting. The short-acting GLP-1 RAs used were exenatide and lixisenatide, and the long-acting ones included liraglutide, dulaglutide and semaglutide. We have now added this information to the "Methods" section of our manuscript (P.11-12).

In the discussion, we have elaborated on how the differences in the pharmacokinetic profiles between short-acting and long-acting GLP-1 RAs can lead to varied pharmacodynamic effects. For instance, short-acting compounds primarily affect postprandial glucose levels by slowing gastric emptying and inhibiting post-meal glucagon release, while long-acting compounds provide a more consistent stimulation of insulin release and suppression of glucagon, which can be beneficial in reducing fasting glucose levels and may result in less tachyphylaxis over time. We have also included the reference to the ELIXA trial (Muskiel, Lancet DE 2018, PMID: 30292589), which provides evidence of the kidney benefits of the short-acting GLP-1 RAs lixisenatide. This trial's findings support the potential for differential effects of GLP-1 RAs based on their duration of action, not only on glycemic control but also on kidney outcomes. We believe that these additions will greatly enhance the discussion of our

results, providing a more complete picture of how the pharmacokinetic properties of GLP-1 RAs may lead to specific clinical benefits in patients with type 2 diabetes and AKD.

We appreciate your valuable suggestions and have incorporated these, especially lixisenatide in ELIXA into our manuscript accordingly (P.22-23).

5. Methods (Page 9; prespecified outcomes): Was mortality in the MAKE and MACE defined as all-cause mortality, or death due to renal and cardiovascular disease respectively?

Response: Thank you for your thorough review and insightful query regarding the mortality definitions in MAKE and MACE as described in our manuscript. To clarify, within the context of our study, MACE indeed encompasses mortality attributed to cardiac death (ICD, I46), as explicitly stated in our text. However, your question regarding the specificity of the mortality causes within the MAKE definition highlights a significant methodological constraint in our research.

Due to the nature of our data collection, which relies on diagnostic codes, we acknowledge that it is not feasible to precisely determine whether mortality was directly related to renal causes. This limitation stems from the inherent restrictions of using diagnostic codes for mortality attribution, which may not always allow for a clear distinction between renal-related and other causes of death.

We are grateful for your insightful comments and the opportunity to clarify this aspect of our study. We have revised the “Discussion section”

(P.25), especially study limitation. Thank you once again for contributing to the improvement of our manuscript.

6. Methods (Page 10, Study Cohort, Line 158): Please rephrase the definition of the index date; it seems it was 90 days following hospital discharge; this marks an important factor of the study.

Response: Thank you for your valuable suggestion, we have revised the sentences accordingly (P.10).

7. Results (Page 17): Do the authors have an explanation why the negative outcomes all seem to be numerically higher in the GLP-1RA group, with some even almost reaching significance? Note that Crohn's disease is spelled incorrectly in the text and supplementary table.

Response: We sincerely appreciate your detailed examination of our manuscript and the valuable comments you have provided, especially regarding the observed results of negative outcomes and the spelling error identified.

Regarding the negative outcomes observed in the GLP-1 RAs group, we have considered several potential explanations for the numerically higher incidence of negative outcomes compared to the control group. These include, but are not limited to, the baseline characteristics of the patient population, the inherent pharmacological effects of GLP-1 RAs,

and possible unmeasured confounders that could influence the outcomes. It is important to note that our study was designed to explore associations rather than establish causality, and these findings suggest avenues for further investigation rather than definitive conclusions about the safety profile of GLP-1 RAs.

In our analysis, we adjusted for known confounders to the extent possible; however, as with any observational study, the potential for residual confounding exists. Additionally, the patient population receiving GLP-1 RAs may have had a higher baseline risk for certain adverse outcomes, which could not be fully accounted for in our analysis. It should be noted that despite the numerically higher incidence of negative outcomes in the GLP-1 RAs group, the overall prognosis may still be favorable in those treated with GLP-1 RAs. This observation could reflect the higher baseline risk for these outcomes among patients treated with GLP-1 RAs.

Thank you once again for your insightful feedback and for contributing to the improvement of our work.

8. Discussion (Page 21, line 338): Please show data that indicate that GLP-1RA's reduce hyperfiltration, as this -as far as I'm aware- has not been shown in eGFR trajectories of large outcome trials (while it is seen in those studying SGLT2 inhibitors), and specific mechanistic studies in humans did not find a beneficial effect of these drugs on measured (intra)renal hemodynamics.

Response: Thank you for your valuable comments on the renoprotective effects of GLP-1 RAs. Recently, in individuals with type 2 diabetes and CKD, the FLOW trial indicates a deceleration in CKD progression and a reduction in kidney and CV mortality risk [1].

In our study, we noted a more gradual eGFR decline among users of GLP-1 RAs, which may hint at an influence on renal hemodynamics. The suggestion that reductions in glomerular hyperfiltration could serve as a renoprotective mechanism was informed by established hypotheses [2-3], yet it was not a direct finding from our research.

In response to your insights, we have revised our discussion for clarity. We now state more cautiously that our study suggests a potential mechanism involving the modulation of renal hemodynamics by GLP-1 RAs, recognizing that this concept has not been definitively proven. However, as a retrospective analysis in nature, we acknowledge and emphasize that our study does not provide direct evidence of changes in glomerular hyperfiltration, and we agree that the eGFR trends observed warrant further investigation to uncover the precise underlying mechanisms.

We are grateful for the chance to refine our manuscript (P.21-22) to better align with current scientific knowledge and are eager to contribute to the ongoing dialogue on this important clinical topic.

[Reference]

1. Gragnano, F., De Sio, V. & Calabrò, P. FLOW trial stopped early due to evidence of renal protection with semaglutide. *Eur Heart J Cardiovasc Pharmacother* **10**, 7-9 (2024).
2. Tonneijck, L., *et al.* Glomerular Hyperfiltration in Diabetes: Mechanisms, Clinical Significance, and Treatment. *J Am Soc Nephrol* **28**, 1023-1039 (2017).

3. Muskiet, M.H.A., *et al.* GLP-1 and the kidney: from physiology to pharmacology and outcomes in diabetes. *Nat Rev Nephrol* **13**, 605-628 (2017).

Thank you once again for your thorough consideration of our study. We firmly believe that the revisions made in response to the Reviewers' valuable feedback have significantly improved the clarity and quality of our research findings. We are hopeful that you will find our manuscript suitable for publication.

In addition to addressing these suggestions, we have also enlisted the assistance of one author to enhance our response to the queries:

1. **Tao-Min Huang:** Tao-Min Huang has been included as an author in this revision due to his significant contributions to data analysis and data interpretation, which have notably improved the overall quality of our research. His expertise in clinical nephrology has played a pivotal role in advancing our work.

We have confirmed that all authors have reviewed and agreed to these changes in authorship. We remain committed to the accuracy and integrity of our manuscript. This addition of author was made to further strengthen the research, and we appreciate your understanding of this change.

REVIEWERS' COMMENTS

Reviewer #1 (Remarks to the Author):

I am fully satisfied about the revision. The authors did a good job.

Reviewer #2 (Remarks to the Author):

I thank you for your time and effort to extensively address all queries raised in your robust point-by-point response. I believe the added analyses and your thorough revision of the manuscript greatly enhanced its scientific quality, balance and clinical relevance. I have no further comments.